# Cold-Azurin, a New Antibiofilm Protein Produced by the Antarctic Marine Bacterium *Pseudomonas* sp. TAE6080

**DOI:** 10.3390/md22020061

**Published:** 2024-01-25

**Authors:** Caterina D’Angelo, Marika Trecca, Andrea Carpentieri, Marco Artini, Laura Selan, Maria Luisa Tutino, Rosanna Papa, Ermenegilda Parrilli

**Affiliations:** 1Department of Chemical Sciences, University of Naples “Federico II”, Complesso Universitario Monte S. Angelo, Via Cintia 4, 80126 Naples, Italy; caterina.dangelo@unina.it (C.D.); andrea.carpentieri@unina.it (A.C.); tutino@unina.it (M.L.T.); 2Department of Public Health and Infectious Diseases, Sapienza University, Piazzale Aldo Moro 5, 00185 Rome, Italy; marika.trecca@uniroma1.it (M.T.); marco.artini@uniroma1.it (M.A.); laura.selan@uniroma1.it (L.S.); rosanna.papa@uniroma1.it (R.P.)

**Keywords:** *Staphylococcus epidermidis*, antibiofilm, azurin, cold-adapted bacteria

## Abstract

Biofilm is accountable for nosocomial infections and chronic illness, making it a serious economic and public health problem. *Staphylococcus epidermidis*, thanks to its ability to form biofilm and colonize biomaterials, represents the most frequent causative agent involved in biofilm-associated infections of medical devices. Therefore, the research of new molecules able to interfere with *S. epidermidis* biofilm formation has a remarkable interest. In the present work, the attention was focused on *Pseudomonas* sp. TAE6080, an Antarctic marine bacterium able to produce and secrete an effective antibiofilm compound. The molecule responsible for this activity was purified by an activity-guided approach and identified by LC-MS/MS. Results indicated the active protein was a periplasmic protein similar to the *Pseudomonas aeruginosa* PAO1 azurin, named cold-azurin. The cold-azurin was recombinantly produced in *E. coli* and purified. The recombinant protein was able to impair *S. epidermidis* attachment to the polystyrene surface and effectively prevent biofilm formation.

## 1. Introduction

Current advances in surgical and medical science are increasing the use of indwelling medical devices; however, their implantation and in vivo use are compromised by their susceptibility to microbial colonization causing half of the problems in healthcare-associated infections (HAIs) [1,2,3]. Their surface and long-term use can be threatened by the adhesion and proliferation of microorganisms, which can interact and form biofilms exposing the body to a risk of permanent colonization and device potential replacement [4]. Staphylococci are the most common etiological agents of medical device infections in the US and Europe [5]. Among them, *Staphylococcus epidermidis* represents one of the leading species of contamination, due to its normal presence on human skin; it can colonize numerous sites throughout the body, and it is now the most common non-aureus Staphylococci (NAS) species associated with infections of indwelling medical devices, endocarditis, and neonatal infections [6]. The ability of *S. epidermidis* to form biofilm on indwelling medical devices is central in the biomedical infection process and represents its major virulence determinant, rendering *S. epidermidis* a successful nosocomial pathogen stem [7]. It is noteworthy that currently, more than 70% of *S. epidermidis* healthcare-associated infections are methicillin-resistant [8] which is of concern and limits the options for effective antimicrobial treatment.

Currently, the treatment of staphylococcal biofilm infections is an expensive and significant challenge, and the phenotypic heterogeneity within the biofilm population associated with the reduced antibiotic susceptibility is a major obstacle for successful antimicrobial therapy. It is established that the biofilm confers competitive advantages to staphylococci, including a 10 to 1000-fold increase in resistance to antibiotics compared to planktonic cells [9]. The extracellular polymeric matrix of biofilm (EPS) protects the embedded bacteria hindering antibiotic penetration and makes it difficult for the drug to reach the inner layer of the biofilm [10]. Additionally, the presence in biofilm of cells with different metabolic features promotes the emergence of tolerant and persister cells which exhibit physiological characteristics that make *S. epidermidis* cells resilient to high antibiotic concentrations [5]. Notably, extrinsic factors also play a role in the modulation of antibiotic susceptibility in staphylococci; for example, mixed biofilms have been shown to confer to *S. epidermidis* an enhanced tolerance toward vancomycin and other antimicrobials [11]. As a result, antimicrobial therapy often fails, and innovative approaches are needed to prevent bacterial adhesion and biofilm formation in the medical setting. The antibiofilm strategies do not aim to inhibit bacterial growth and cell division, but instead target molecules and pathways involved in the formation and maturation of biofilms without necessarily killing biofilm-associated cells. This latter should allow the development of narrow-spectrum agents, which will have low or no influence on commensal microbiota [3]; moreover, an approach that targets biofilm without affecting bacterial vitality avoids the rapid appearance of escape mutants. Therefore, the use of antibiofilm molecules that are active against different stages of biofilm development represents a promising option.

Microorganisms able to survive in extreme environments, such as Antarctica, can be a promising source of new antibiofilm agents. Antarctic marine bacteria apply various survival strategies to persist in harsh conditions, and the reduction of competing microorganisms is one of these approaches. Due to the key role of biofilm in bacteria fitness in extreme conditions [12], the production of antibiofilm molecules could reduce the competitors’ survival. Indeed, several papers report that marine Antarctic bacteria produce and secrete antibiofilm molecules [13,14,15,16,17,18].

In this work, we investigated the ability of the Antarctic marine bacterium *Pseudomonas* sp. TAE6080 [19] to produce compounds effective against *S. epidermidis* biofilm formation and we identified the molecule responsible for this activity.

The antibiofilm molecule is a protein named cold-azurin. The results reported in this paper sustain once again the great potential of Antarctic bacteria as producers of bioactive molecules.

## 2. Results

### 2.1. Production of AntiBiofilm and Anti-Adhesive Molecules Active against S. epidermidis Biofilm

*Pseudomonas* sp. TAE6080 is able to produce and secrete molecule/s able to interfere with the biofilm formation of *S. epidermidis* RP62A [19]. Firstly, the ability of this Polar bacterium to also produce anti-adhesive compound was evaluated. The anti-adhesive and antibiofilm activities of *Pseudomonas* sp. TAE6080 cell-free supernatant (SN TAE6080) were tested against the reference strain *S. epidermidis* RP62A [19] and against *S. epidermidis* O-47 [20], an agr-mutant considered a strong biofilm producer [21].

SN TAE6080′s anti-adhesive capability was tested using the surface coating assay by evaluating the ability of the coated surfaces to avoid staphylococcal biofilm formation (Figure 1A). The capability of SN TAE6080 to inhibit the biofilm formation of *S. epidermidis* was evaluated by adding the supernatant to the staphylococcal growth medium at the beginning of the cultivation (Figure 1B). As reported in Figure 1A, SN TAE6080 showed anti-adhesive activity against both *S. epidermidis* strains; moreover it was able to impair the biofilm formation of the two strains (Figure 1B).

To ascertain more information about the chemical nature of the anti-adhesive and/or antibiofilm compound/s, SN TAE6080 was treated with proteinase K, and the activity of the treated samples was compared to the activity of untreated samples. The treatments affected both anti-adhesive (Figure 1C) and biofilm inhibiting activity (Figure 1D), suggesting that the activities could be due to a peptide and/or proteins. To have more information about the molecular weight of the active compound/s, the SN TAE6080 was concentrated ten-fold by ultrafiltration with membranes of 30 kDa MWCO, and the anti-adhesive and antibiofilm activity of the retentate fraction (SNC) and the permeate fraction was determined by the coating assay against *S. epidermidis* RP62A and *S. epidermidis* O-47. Moreover, SNC can inhibit the initial attachment and the biofilm formation (data not shown).

### 2.2. Purification of Anti-Adhesive and Antibiofilm Molecule

To have an amount of cell-free supernatant sufficient for the purification, a scale-up of *Pseudomonas* sp. TAE6080 growth in a bioreactor was set up. The Antarctic bacterium was grown in a 3 L stirred tank fermenter at 15 °C in synthetic G medium. The growth was followed for 72 h till the stationary phase, and the supernatant (SN TAE6080) was collected, separated from the cells, and ultrafiltered with membranes of 30 kDa MWCO as described above. To purify the active molecule, SNC was subjected to adsorption chromatography on Amberlite XAD-2, a polystyrene resin, exploiting the reported ability of the anti-adhesive molecules to adhere to this material. Upon SNC loading, the column was extensively washed with G medium, while the elution steps were performed with methanol. The collected chromatographic fractions were tested by the surface coating assay to evaluate their anti-adhesive properties against *S. epidermidis* RP62A and O-47 (Figure 2A). As shown in Figure 2A, the fraction eluted with methanol (named E), dried and resuspended in G medium, displays a strong antibiofilm activity by interfering with *S. epidermidis* adhesion on the entire well surface. This activity was confirmed also by biofilm inhibiting assay against both *S. epidermidis* strains (Figure 2B). Given the putative proteinaceous nature of the anti-adhesive molecule/s, the fractions were analyzed by SDS-PAGE. The electrophoresis analysis demonstrated that the eluted fraction (E) had a less complex protein profile compared to the unbound (Ub) and wash fraction (W), and that the E fraction was enriched in two proteins (Figure 2C).

### 2.3. Cold-Azurin Identification

SDS-PAGE analysis of the active fraction obtained after the proposed purification protocol showed the presence of two main protein bands, which were subsequently in situ hydrolyzed and peptide mixtures thus obtained were analyzed by LC-MS/MS for proteins identifications as described in the Section 4. Our results led to the identification of two *Pseudomonas* sp. TAE6080 proteins namely a flagellin [22] (FliC the filament core of flagellum) with a molecular weight of about 60 kDa (Figure 2C) and a 15 kDa (Figure 2C) protein displaying a good similarity (81.8%) to the azurin [23] from Pseudomonas aeruginosa PAO1 (Appendix A); both *Pseudomonas* sp. TAE6080 proteins were identified with a 35% of sequence coverage (Figure 2D,E). The two proteins present in the active fraction and identified by MS usually have a different cell localization. Indeed, azurin is a periplasmic protein while flagellin is a secreted protein. We took advantage of this difference to understand which protein was responsible for antibiofilm and anti-adhesive activity.

Given the periplasmic nature of the azurin, the outer membrane permeabilization by cold osmotic shock was performed on TAE6080 cell-pellet. The TAE6080 periplasmic extract named for simplicity OS, presumably enriched with cold-zurin (Figure 3A), was analyzed for anti-adhesive and antibiofilm activity (Figure 3B,C) against *S. epidermidis* strains and proved to be able to impair the biofilm formation (Figure 3C) of both staphylococci and to have strong anti-adhesive activity (Figure 3B). *Pseudomonas* sp. TAE6080 periplasmic extract (OS) was more active against *S.epidermidis* RP62A while the cell-free supernatant was more efficient against *S.epidermidis* O-47 biofilm. The presence of some *Pseudomonas* sp. TAE6080 periplasmic proteins could be responsible for the recorded different behavior; future experiments will be dedicated to clarifying this point.

### 2.4. Recombinant Cold-Azurin Production and Purification

Although the antibiofilm and anti-adhesive activity of TAE6080 periplasmic extract (OS) is a strong clue that the protein responsible for the reported activity is the cold-azurin, we produced the cold-adapted protein in *E. coli* cells to confirm our results. As reported in Section 4, the cold-azurin gene was cloned into pET28b(+) vector under the control of an inducible lac operon using *E. coli* BL21DE3 as a host for the heterologous production. We chose to express the protein at 28 °C to not eventually compromise the correct folding of the rcold-azurin. Appendix A reports the electrophoretic profiles of the periplasmic fractions of recombinant cells induced or non-induced with IPTG. Rcold-azurin was produced in a very large amount only when the inducer was added to the medium (+IPTG). The E. coli periplasmic extract containing the recombinant rcold-azurin was biologically active as demonstrated by the coating assay on *S. epidermidis* O-47 strain and by biofilm inhibiting assay (Appendix A).

To purify the rcold-azurin, the same protocol used for the purification of native protein was applied. The periplasmic fraction from recombinant *E. coli* BL21DE3 induced cells was subjected to adsorption chromatography (Amberlite XAD-2) and the elution step was performed with methanol. After the dry, the eluted fraction was suspended in a small volume of PBS and the chromatography fractions were analyzed by SDS-PAGE, demonstrating the presence of only rcold-azurin in the eluted sample (Figure 4A). The purified rcold-azurin activity was evaluated and the recombinant protein showed strong antibiofilm (Figure 4B) and anti-adhesive capability (Figure 4C) against both *S. epidermidis* strains.

The effect of rcold-azurin on both *S. epidermidis* strains biofilms was further explored by confocal laser scanning microscopy (CLSM). The biofilm structure and cell integrity were analyzed using a LIVE/DEAD^®^ Biofilm Viability Kit. As shown (Figure 5A), the CLMS analysis confirmed that rcold-azurin can reduce *S. epidermidis* biofilm formation without affecting cell viability. The CLSM image stack data on treated biofilms were analyzed using the COMSTAT 2 image analysis software package [24] to evaluate the different variables describing the biofilm structure. As expected, the values of the biomass and the average thickness of the biofilm obtained in the presence of rcold-azurin were lower if compared with the values obtained without the antibiofilm protein, while an increased roughness coefficient is observed for the treated sample (Figure 5B). This dimensionless factor grants a measure of the thickness variation of a biofilm, and thus it is used as an indirect indicator of biofilm heterogeneity. The assays revealed that the treatment resulted in an unstructured and non-homogeneous biofilm compared to untreated biofilm.

## 3. Discussion

In a previously published paper [19], we reported the ability of *Pseudomonas* sp. TAE6080 to produce antibiofilm molecule/s capable of inhibiting the formation of *S. epidermidis* RP62A biofilm. In this paper, we demonstrated that the bacterium produces and secretes molecule/s able to impair the biofilm formation of another *S. epidermidis* strain, *S. epidermidis* O-47, and to interfere also with the attachment to the polystyrene surface of both strains. The observation that *Pseudomonas* sp. TAE6080 cell-free supernatant interfered with the surface adhesion and with biofilm formation suggested the presence of two different molecules involved in, or the presence of a single molecule capable of, acting as an anti-adhesive and antibiofilm compound. To clarify this point, and to collect physico-chemical information, the cell-free supernatant was treated with proteinase K or subjected to ultrafiltration and then tested against the two pathogens. The results demonstrated the involvement of a protein/s with molecular weight higher than 30 kDa. The purification strategy for the *Pseudomonas* sp. TAE6080 anti-adhesive molecule/s was based on the ability of the protein/s to bind polystyrene. As expected, the fraction eluted from the polystyrene chromatographic resin with methanol (E) had the ability to interfere with *S. epidermidis* adhesion and, surprisingly, it presented good antibiofilm activity. These results indicated that the bacterium produces protein/s that impairs *S. epidermidis* attachment to the polystyrene surface and effectively prevents biofilm formation. The E fraction analysis by SDS-PAGE revealed the presence of two proteins identified by LC-MS/MS, a flagellin and a 15 kDa protein similar (81.8% of similarity) to the azurin [25] from *P. aeruginosa* PAO1 that was named cold-azurin.

Several proteins have emerged as good candidates for biofilm treatment and prevention [26,27] but are generally hydrolytic enzymes. In our case, the two potential antibiofilm proteins are known to be involved in different biological processes. The flagellin protomers are secreted by the flagellar secretion apparatus and are arranged as a multimer to form a long filament [22]. Azurin is a low molecular weight, blue, copper-containing protein found mainly in the periplasmic space of various Gram-negative bacteria [28]. A possible antibiofilm activity of flagellin seems to be counterintuitive and the involvement of azurin in the prevention of biofilm is not so obvious. Therefore, we awarded the antibiofilm activity to the cold-azurin by testing the activity of the periplasmic extract of *Pseudomonas* sp. TAE6080 and evaluating the activity of the recombinant cold-azurin produced in *E. coli*. Both samples were active against *S. epidermidis* biofilm formation and displayed good anti-adhesive capability. Moreover, the CLSM analyses on *S. epidermidis* treated biofilm revealed that the purified *r*cold-azurin not only reduced the biofilm biomass but deeply modified the *S. epidermidis* biofilm structure without affecting cell viability.

These results allowed us to attribute the biological activity of the *Pseudomonas* sp. TAE6080 cell-free supernatant to the cold-azurin.

Azurin acts in a respiratory electron transport chain in some bacteria, and in *Pseudomonas aeruginosa* PAO1 it is involved in denitrification and protection against oxidative stress [23,29,30]. Indeed, azurin allows single-electron transfer between enzymes associated with the cytochrome chain by undergoing oxidation-reduction between Cu(I) and Cu(II), and also supports oxidative deamination of primary amines by transferring electrons from aromatic amine dehydrogenase to cytochrome oxidase, as well as from some c-type cytochromes to nitrite reductases [25]. In an attempt to speculate that the antibiofilm activity of cold-azurin might be related to the Cu(2+)/Cu(+) reduction potentials of the type-1 copper site [31], we tested the putative antibiofilm activity of poxA1b laccase [32], a well-known blue copper protein oxidase (Appendix A). The obtained results showed the absence of antibiofilm activity due to the laccase; thus, the molecular mechanism responsible for the activity of cold-azurin is not related to its redox property.

Unrelated to its electron-transfer property, azurin has been found to be active against different agents of human diseases such as malaria, AIDS, and cancer [33,34]. Moreover, this protein inhibits the attachment and invasion of different pathogenic bacteria to host cells [35]. This versatile protein is able to interact with unrelated targets such as the surface protein MSP1-19 [36] of the malarial parasite *Plasmodium falciparum*, the HIV-protein gp120 [36], the ephrin receptor EphB2 [37], and the tumor suppressor protein p53 [38].

Fialho and coworkers suggested that the azurin’s promiscuity in targeting multiple proteins is related to its three-dimensional structure [33]. Azurin is a member of the family cupredoxins [37] and members of this family demonstrate structural features similar to the immunoglobulin variable domains [33,39]. Indeed, although azurin and the immunoglobulins have a low sequence identity, their similarity is based on the presence of invariant super secondary substructures common to cupredoxins and immunoglobulins [33]. The hypothesis proposed by Fialho and coworkers [33] is that azurin is used by the bacterium as a multitarget weapon to avoid the entry of pathogenic competitors into the host cell and to eliminate foreign invaders from the host organism. In this way, the bacterium preserves its own survival. This behavior is exactly the job of the immune system, which is made up of immunoglobulins.

In this view, the reported antibiofilm activity of cold-azurin can be interpreted as an additional strategy to reduce the presence of potential competitors and the activity could be related to the ability of cold-azurin to interact with specific proteins required for biofilm formation. Several *S. epidermidis* proteins play an important role in biofilm formation [38,39,40,41]. For example, the A domain of the accumulation associated protein (Aap) can promote adhesion to unconditioned biomaterial [42,43], small basic proteins (Sbp) can foster *S. epidermidis* biofilm formation [44], and the surface protein AtlE [45], a bi-functional adhesin/autolysin abundant in the cell wall of *S. epidermidis*, has a key role in *S. epidermidis* biofilm formation [41]. Therefore, further experiments are required to assess whether the target of cold-azurin activity is one of the proteins involved in *S. epidermidis* biofilm development. In addition, future studies will aim for the complete biochemical characterization of the cold-azurin and its comparison with the extensively studied azurin from *P. aeruginosa* PAO1. However, the present study paves the way for the use of cold-azurin as a potential agent against *S. epidermidis* biofilm.

Furthermore, the studies on the potential use of azurin from *P. aeruginosa* in various human pathologies [33,35,46,47], which also demonstrated its biocompatibility [34,48,49], are encouraging for the future use of cold-azurin in the treatment of human infections in combination with conventional antibiotics.

## 4. Materials and Methods

### 4.1. Bacterial Strains and Culture Conditions

Bacterial strains used in this work were *Pseudomonas* sp. TAE6080, collected in 1992 from seawater near French Antarctic Station Dumont d’Urville, Terre Adélie (66°40′ S; 140°01′ E) [19]; *S. epidermidis* O-47 [21] isolated from clinical septic arthritis; *S. epidermidis* RP62A reference strain [50] isolated from an infected catheter (ATCC collection no. 35984).

*Pseudomonas* sp. TAE6080 was grown in synthetic medium G (D-Gluconic acid sodium 10 g L^−1^, NaCl 10 g L^−1^; NH_4_NO_3_ 1 g L^−1^; KH_2_PO·7H_2_O 1 g L^−1^; MgSO_4_·7H_2_O 200 mg L^−1^; FeSO_4_·7H_2_O 5 mg L^−1^; CaCl_2_·2H_2_O 5 mg L^−1^) [19] in planktonic conditions at 15 °C under vigorous agitation (250 rpm) for 72 h of growth. The cell-free supernatant was separated from the pellet by centrifugation (7000 rpm at 4 °C for 30 min), sterilized by filtration through membranes with a pore diameter of 0.22 μm, and stored at 4 °C until use. The cell pellet was stored at –20 °C until use.

Staphylococci were grown at 37 °C in Brain Heart Infusion broth (BHI, Oxoid, Basingstoke, UK), and biofilm formation was assessed in static conditions while planktonic cultures were performed under agitation (180 rpm).

All strains were maintained at −80 °C in cryovials with 20% glycerol.

### 4.2. TAE6080 Cell-Free Supernatant Preparation

The cell-free supernatant (SN TAE6080) was concentrated 10-fold with Amicon Ultrafiltration cell equipped with a 30 kDa cut-off PES Millipore Ultrafiltration Disc (Merck KGaA, Darmstadt, Germany). Then, retentate fraction (SNC) was collected.

### 4.3. Surface Coating Assay

For the surface coating assay [17], a volume of 5 μL of the tested sample was deposited onto the center of a well of a 24-well tissue-culture-treated polystyrene microtiter plate. The plate was incubated at room temperature to allow complete evaporation of the liquid in sterile conditions. The wells were then filled with 1 mL of *S. epidermidis* RP62A or *S. epidermidis* O-47 cultures in exponential growth phase diluted in BHI with a final concentration of about 0.1 and 0.001 OD_600nm_, respectively, and incubated at 37 °C in static condition. After 24 h, wells were rinsed with water and stained with 1 mL of 0.1% crystal violet. Stained biofilms were rinsed with water and dried; after that the wells were photographed.

### 4.4. Biofilm Inhibiting Assay

The quantification of in vitro biofilm production was based on the method described by Ricciardelli and coworkers [51]. For staphylococcal biofilm formation in the presence of *Pseudomonas* sp. TAE6080 cell-free supernatant (SN TAE6080), the wells of a sterile 96-well flat-bottomed polystyrene plate were filled with *S. epidermidis* RP62A or *S. epidermidis* O-47 cultures in exponential growth phase diluted in BHI 2× with a final concentration of about 0.1 and 0.001 OD_600nm_, respectively. Each well was filled with 100 μL of bacterial cultures and 100 μL of the cell-free supernatant. In this way, the sample was used diluted 1:2 with a final concentration of 50%. As a control, the first row was filled with 100 μL of bacterial cultures and 100 μL of G medium (untreated bacteria). The plates were incubated aerobically for 24 h at 37 °C. Biofilm formation was measured using crystal violet staining. After incubation, planktonic cells were gently removed; and wells were washed three times with sterile PBS and thoroughly dried. Each well was then stained with 0.1% crystal violet and incubated for 15 min at room temperature, rinsed twice with double-distilled water, and thoroughly dried. The dye bound to adherent cells was solubilized with 20% (*v*/*v*) glacial acetic acid and 80% (*v*/*v*) ethanol. After 30 min of incubation at room temperature, the OD_590nm_ was measured to quantify the total biomass of biofilm formed in each well. Each data point was composed of six independent samples.

SN TAE6080 was subjected to proteinase K treatment. The antibiofilm activity of treated and untreated supernatant was evaluated using the microtiter plate assay against *S. epidermidis* strains as previously described. Each data point was composed of five independent samples.

For the assay in the presence of *Pseudomonas* sp. TAE6080 periplasmic extract by osmotic shock method (named OS), the wells of a sterile 96-well flat-bottomed polystyrene plate were filled with *S. epidermidis* RP62A or *S. epidermidis* O-47 cultures in exponential growth phase diluted in BHI with a final concentration of about 0.1 and 0.001 OD_600nm_, respectively. Each well was filled with 180 μL of cultures and 20 μL of periplasmic extract. In this way, the sample was used diluted 1:10 with a final concentration of 10%. As a control, the first row was filled with 180 μL of cultures and 20 μL of buffer used for the periplasmic protein extraction (untreated bacteria). The plates were incubated aerobically for 24 h at 37 °C. Biofilm formation was measured as previously described. Each data point was composed of four independent samples.

For the assay with *r*cold-azurin: the wells of a sterile 96-well flat-bottomed polystyrene plate were filled with 200 μL of *S. epidermidis* RP62A or *S. epidermidis* O-47 cultures in exponential growth phase diluted in BHI with a final concentration of about 0.1 and 0.001 OD_600nm_, respectively. The plate was incubated at 37 °C for 24 h in the absence and in the presence of *r*cold-azurin (0.25 µg µL^−1^). Biofilm formation was measured as previously described. Each data point was composed of four independent samples.

### 4.5. Proteinase K Treatment

To analyze the proteinaceous nature of active compound/s, proteinase K (Sigma Aldrich, St Louis, MO, USA) was added to the sample at a final concentration of 2 mg mL^−1^ and the reaction was incubated for 2 h at 37 °C [14]. To exclude an effect due to the temperature, the sample was incubated without proteinase K for 2 h at 37 °C.

### 4.6. Large-Scale Growth

*Pseudomonas* sp. TAE6080 bacterial culture was grown in G medium in a stirred tank reactor 3 L fermenter (Applikon, Schiedam, The Netherlands) connected to an ADI-1030 Bio Controller with a working volume of 1 L. The bioreactor was equipped with the standard pH, pO_2_, level- and temperature sensors for bioprocess monitoring. The culture was carried out at 15 °C for 72 h in aerobic conditions (30% dissolved oxygen). Supernatant was recovered by centrifugation at 7000 rpm. Then, it was sterilized by filtration through membranes with a pore diameter of 0.22 µm and stored at 4 °C until use.

### 4.7. Adsorption Chromatography

The primary enrichment of the active compound/s was achieved by adsorption chromatography on a polystyrene resin (Amberlite XAD-2; Rohm and Haas, Philadelphia, PA, USA). The resin (3 g) was placed in a glass column (10 cm by 1 cm). The column was equilibrated with G medium and then 10 mL of retentate fraction (SNC) was applied at a flow rate of approximately 1 mL min^−1^. The column was then washed with 3-bed volumes of G medium. The elution of the active compound/s was subsequently carried out with methanol. Fractions of 15 mL volume were collected. The fraction eluted with methanol was recovered, dried to obtain more concentrated samples, and resuspended in a small volume of G medium. Each chromatographic fraction was analyzed by surface coating assay against *S. epidermidis* O-47 and *S. epidermidis* RP62A.

For the *r*cold-azurin purification with Amberlite XAD-2, the resin (8 g) was placed in a polypropylene column (12 cm by 1.5 cm) Econo-Pac^®^. The column was equilibrated with 5 mM MgSO_4_ and then 5 mL of periplasmic extract from recombinant BL21DE3 induced cells was applied at a flow rate of approximately 1 mL min^−1^. The column was then washed with 3-bed volumes of 5 mM MgSO_4_. The elution of the *r*cold-azurin was subsequently carried out with methanol. Fractions of 10 mL volume were collected. The fraction eluted with methanol was recovered, dried to obtain more concentrated samples, and resuspended in a small volume (about 0.3 mL) of PBS buffer. The activity of purified protein was analyzed by surface coating assay or antibiofilm assay against *S. epidermidis* O-47 and *S. epidermidis* RP62A.

### 4.8. SDS-PAGE

Protein samples (prepared in Laemmli buffer 4x followed by boiling at 95 °C) were separated on SDS-PAGE gels. The gels were stained with colloidal Coomassie or silver nitrate staining; the protein sizes were determined by comparing the migration of the protein band to a molecular mass standard (Unstained Protein Molecular Weight Marker, Thermo Fisher Scientific, Waltham, MA, USA). To analyze total proteins (tot), 1 OD_600nm_ of liquid cultures were harvested at the end of the growth, centrifuged at 13,000 rpm for 10 min at 4 °C, and the pellet (about 0.75 mg) was solubilized in 60 μL of Laemmli Sample buffer 4×. Then, the sample was boiled at 95 °C for 20 min, quickly cooled on ice for 5 min, and finally centrifuged at 13,000 rpm for 5 min at RT. An aliquot of 2 μL was analyzed by SDS-PAGE.

### 4.9. In Situ Hydrolysis, LC-MS/MS Analysis and Protein Identification

Mono-dimensional SDS-PAGE gel was stained with Coomassie Brilliant Blue, the band approximately at 60 kDa and 15 kDa were excised and de-stained with 100 µL of 0.1 M ammonium bicarbonate (AMBIC) and 130 µL of acetonitrile (ACN) and subsequently subjected to in situ hydrolysis with 0.1 µg µL^−1^ trypsin mM in AMBIC for 18 h at 37 °C. The hydrolysis was stopped by adding acetonitrile and 0.1% formic acid. The sample was then filtered and dried in a vacuum centrifuge.

The peptide mixtures thus obtained were directly analyzed by LTQ Orbitrap XL™ Hybrid Ion Trap-Orbitrap Mass Spectrometer (Thermo Fisher Scientific, Bremen, Germany). C-18 reverse phase capillary column 75 μm × 10 cm (Thermo Fisher Scientific) was performed using a flow rate of 300 nL min^−1^, with a gradient from eluent A (0.2% formic acid in 2% acetonitrile) to eluent B (0.2% formic acid in 95% acetonitrile). The following gradient conditions were used: t = 0 min, 5% solvent B; t = 10 min, 5% solvent B; t = 90 min, 50% solvent B; t = 100 min, 80% solvent B; t = 105 min, 100% solvent B; t = 115 min, 100% solvent B; and t = 120 min, 5% solvent B. Peptide analysis was performed using the data-dependent acquisition of one MS scan followed by CID fragmentation of the five most abundant ions.

For the MS scans, the scan range was set to 400–1800 *m/z* at a resolution of 60,000, and the automatic gain control (AGC) target was set to 1 × 106. For the MS/MS scans, the resolution was set to 15,000, the AGC target was set to 1 × 105, the precursor isolation width was 2 Da, and the maximum injection time was set to 500 ms. The CID normalized collision energy was 35%. Data were acquired by Xcalibur™ software 4.2 (Thermo Fisher Scientific).

In-house Mascot software (version 2.4.0) was used as a search engine to identify proteins. The TAE6080 (2644 sequences; 878,869 residues) proteins database was used for proteins identification.

The software returns a list of proteins associated with a probability index (score), calculated as −10 × Log P, where P is the probability that the observed event is a random one. Proteins are considered as identified if a minimum number of 2 peptides reach the calculated threshold score.

### 4.10. Pseudomonas sp. TAE6080 Periplasmic Protein Extraction

Osmotic shock. A cell pellet of about 0.3 g was resuspended in 5 mL of 30 mM Tris-HCl, pH 7.8 and 20% sucrose, 1× EDTA-free protease inhibitor (Roche). After 30 min of incubation at room temperature, the cells were centrifuged (7500 rpm for 20 min at 4 °C) and the supernatant containing the highest amount of active protein was carefully recovered and named OS. The pellet was then resuspended in 5 mL of ice-cold 0.5 mM MgCl_2_ and gently shaken for 10 min in an ice bath. The suspension was centrifuged (7500 rpm for 20 min at 4 °C). After centrifugation, the soluble fraction was carefully transferred into a new tube and the pellet, containing the cytoplasmatic fraction, was suspended in 30 mL lysis buffer (50 mM Tris-HCl, pH 7.8, 500 mM NaCl, one tablet of EDTA-free Complete Ultra protease inhibitor (Roche, Mannheim, Germany). The cells were mechanically lysed by a French Press at 2 Kbar for two consecutive cycles. The obtained lysate was centrifuged (6500 rpm for 1 h at 4 °C) to separate the soluble and insoluble protein fractions, then the soluble fraction (Sol) was analyzed on SDS-PAGE.

### 4.11. Recombinant Cold-Azurin Protein Production

For the production of the recombinant protein, the azurin gene was PCR amplified and cloned into the commercial expression vector pET28b (+). The resulting construct was indicated as pET28b-Azu. The azurin gene was amplified by forward primer Azu-NdeI: 5′- CCCTGGATCCGAGATTCATATGTTTGCC -3′ and reverse primer Azu-BamHI: 5′- CGATGAAGGATCCCGCGGTCTTGAG -3′. The primers were designed based on *Pseudomonas* TAE6080 sequence (GenBank, under the accession number JAHIDY000000000) which encompasses the whole sequence of azurin gene and harbors the restriction sites of *Nde*I and *Bam*HI enzymes, respectively. The PCR was performed with initial denaturation (98 °C, 30 s), 25 cycles of denaturation (98 °C, 10 s), annealing (56 °C, 30 s) and extension (72 °C, 30 s), and final extension (72 °C, 2 min). The products were electrophoresed on 1.5% agarose gel, purified using High Pure PCR Product Purification Kit (Roche, Mannheim, Germany) and sequenced.

The cloning of azurin gene in pET28b vector was performed via double digestion of purified PCR product by 20 U µL^−1^ *Bam*HI-*Nde*I, (Thermo Fisher Scientific, New York, NY, USA). The pET8b vector was also double digested with the same restriction enzymes. The digested fragments were purified using High Pure PCR Product Purification kit (Roche, Mannheim, Germany) and incubated at 16 °C overnight with T7 ligation enzyme (Thermo Fisher Scientific, New York, NY, USA). The pET28b-Azu construct was transformed into the *E. coli* BL21DE3 competent cells, plated on Luria-Bertani (LB) agar (Sigma aldrich, Steinheim, Germany) in the presence of 50 µg mL^−1^ of kanamycin (Sigma aldrich, Steinheim, Germany), and incubated overnight at 37 °C.

The transformed *E. coli* BL21DE3 cells containing pET28b-Azu construct were subjected to protein expression and production. Temperature and induction condition for *r*cold-azurin expression were determined and optimized.

Briefly, the transformed colonies were inoculated into 10 mL of LB broth without antibiotic at 28 °C, 180 rpm. After 1 h, 10 mL of inoculum were diluted in 100 mL of fresh LB medium containing 50 µg mL^−1^ of kanamycin and 5 µg ml^−1^ CuSO_4_ until reaching the optical density of 0.5 at 600 nm (about 2 h) at 28 °C, 180 rpm. The expression was induced by the addition of isopropyl-1-thio-β-D-galactopyranoside (IPTG) 2 mM to the culture medium and incubated for 16–20 h. After the incubation, cells were harvested by centrifugation at 4000 rpm at 4 °C.

The bacterial pellet was washed with 150 mM of Ice-cold phosphate buffer at pH 7, resuspended in buffer A (30 mM Tris–HCl pH 8, 20% sucrose, 1 mM EDTA pH 8) and incubated at room temperature for 20 min. The shocked cells were collected by centrifugation at 8000 rpm at 4 °C and resuspended in ice-cold 5 mM MgSO_4_. After incubation at 4 °C for 20 min and centrifugation at 13,000 rpm, the supernatant (periplasmic fractions) containing the recombinant azurin was collected and stored at −20 °C until use. The protein content was determined by Bradford assay.

### 4.12. Confocal Microscopy

The activity of purified *r*cold-azurin against staphylococcal biofilms was evaluated by Confocal Laser Scanning Microscopy (CLSM). Biofilms were formed on NuncTM Lab-Tek^®^ 8-well Chamber Slides (n◦17744; Thermo Scientific, Ottawa, ON, Canada) [15]. Briefly, the wells of the chamber slide were filled with 300 μL of *S. epidermidis* RP62A or *S. epidermidis* O-47 cultures in exponential growth phase diluted in BHI with a final concentration of approximately 0.1 and 0.001 OD_600nm_, respectively. The culture was incubated at 37 °C for 24 h in the absence (control) and in the presence of protein (0.25 µg µL^−1^) to assess its antibiofilm activity and its influence on cell viability. The biofilm cell viability was determined by the FilmTracer™ LIVE/DEAD^®^ Biofilm Viability Kit (Molecular Probes, Invitrogen, Carlsbad, CA, USA), following the manufacturer’s instructions. After rinsing with filter-sterilized PBS, each well of the chamber slide was filled with 300 µL of working solution of fluorescent stains, containing SYTO^®^9 green-fluorescent nucleic acid stain (10 µM) and propidium iodide, the red-fluorescent nucleic acid stain (60 µM), and incubated for 20–30 min at room temperature, protected from light. All excess stain was removed by rinsing gently with filter-sterilized PBS. All microscopic observations and image acquisitions were performed with a confocal laser scanning microscope (LSM700-Zeiss, Jena, Germany) equipped with an Ar laser (488 nm) and a He-Ne laser (555 nm). Images were obtained using a 20×/0.8 objective. The excitation/emission maxima for these dyes are 480/500 nm for SYTO^®^9 and 490/635 nm for PI. Z-stacks were obtained by driving the microscope to a point just out of focus on both the top and bottom of the biofilms. Images were recorded as a series of tif files with a file-depth of 16 bits. The COMSTAT software package [24] was used to determine biomasses (μm^3^ μm^−2^), average thicknesses (µm), and roughness coefficient (Ra*). For each condition, two independent biofilm samples were used.

### 4.13. Statistics and Reproducibility of Results

The significance of differences between the mean values was calculated using a two- tailed Student’s *t*-test and *p* < 0.05 was considered significant.

## Figures and Tables

**Figure 1 marinedrugs-22-00061-f001:**
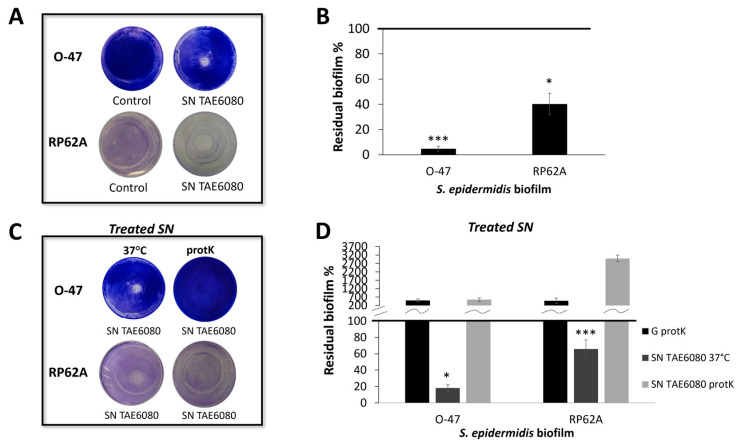
(**A**) Biofilm formation by *S. epidermidis* O-47 and *S. epidermidis* RP62A in polystyrene 24-wells microtiter plate wells coated with G medium (Control) and SN TAE6080 obtained after 72 h of growth. (**B**) Effect of SN TAE6080 on *S. epidermidis* biofilm formation. Each data point represents the mean ± SD of six independent samples. The results are expressed as the percentage of biofilm formed in the presence of SN TAE6080 compared to untreated bacteria (100%). Biofilm formation was considered unaffected in the range of 90–100%. Differences in mean absorbance were compared to the untreated control and considered significant when *p* < 0.05 (* *p* < 0.05, *** *p* < 0.001) according to the Student’s *t*-test. (**C**) Biofilm formation by *S. epidermidis* O-47 and *S. epidermidis* RP62A in polystyrene 24-wells microtiter plate wells coated with SN TAE6080 treated with proteinase K (protK) or SN TAE6080 incubated at 37 °C without proteinase K as a control. (**D**) Effect of SN TAE6080 treated with proteinase K on biofilm formation. As controls, the effect of SN TAE6080 incubated at 37 °C without proteinase K and the effect of G medium treated with proteinase K were reported. Each data point represents the mean ± SD of five independent samples. The results are expressed as the percentage of biofilm formed in the presence of SN TAE6080 compared to untreated bacteria (100%). Biofilm formation was considered unaffected in the range of 90–100%. Differences in mean absorbance were compared to the untreated samples and considered significant when *p* < 0.05 (* *p* < 0.05, *** *p* < 0.001) according to the Student’s *t*-test.

**Figure 2 marinedrugs-22-00061-f002:**
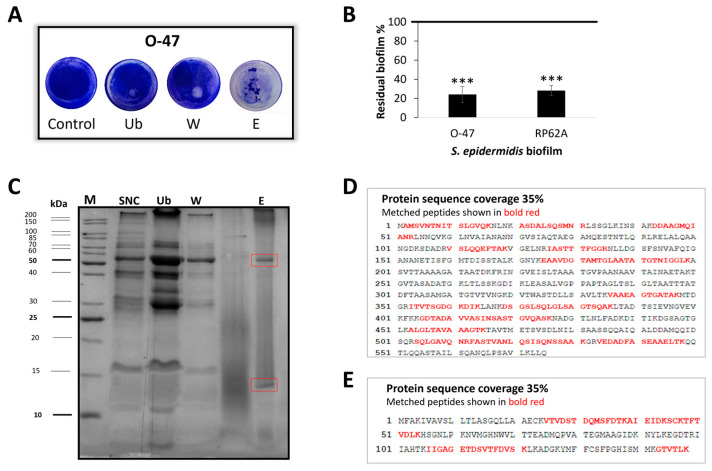
(**A**) Biofilm formation by *S. epidermidis* O-47 in polystyrene 24-wells microtiter plate wells coated with G medium (Control) and fractions obtained from adsorption chromatography: Ub: unbound fraction, W: fraction obtained during the washing step with G medium, E: fraction eluted with methanol. (**B**) Effect on *S. epidermidis* biofilm formation of E fraction eluted with methanol from adsorption chromatography. Each data point represents the mean ± SD of four independent samples. The results are expressed as the percentage of formed biofilm in the presence of E fraction compared to untreated bacteria (100%). Biofilm formation was considered unaffected in the range of 90–100%. Differences in mean absorbance were compared to the untreated control and considered significant when *p* < 0.05 (*** *p* < 0.001) according to the Student’s *t*-test. (**C**) 15% SDS-PAGE stained with Comassie blue, protein profile of the chromatography fractions (Ub, W, E) and retentate fraction of cell-free supernatant (SNC). SNC: loaded sample on chromatography, Ub: unbound fraction, W: fraction obtained during the washing step, E: fraction eluted with methanol, M: molecular weight marker (**D**) Flagellin protein sequence coverage obtained by LCMSMS analysis. (**E**) Azurin protein sequence coverage obtained by LCMSMS analysis.

**Figure 3 marinedrugs-22-00061-f003:**
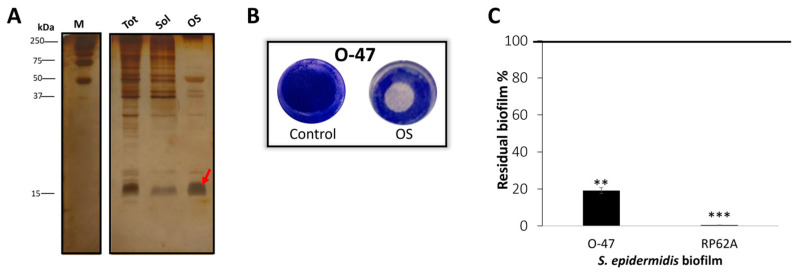
(**A**) SDS-PAGE 15% stained with silver staining, protein profile of cell pellet before osmotic shock (tot), soluble post lysis (Sol), and TAE6080 periplasmic fraction (OS), M: molecular weight marker. (**B**) Biofilm formation by *S. epidermidis* O-47 in polystyrene 24-wells microtiter plate wells coated with OS buffer (Control) or TAE6080 periplasmic fraction (OS). (**C**) Effect of TAE6080 periplasmic fraction (OS) on *S. epidermidis* biofilm formation. Each data point represents the mean ± SD of four independent samples. The results are expressed as the percentage of biofilm formed in the presence of TAE6080 periplasmic fraction (OS) compared to untreated bacteria (100%). Biofilm formation was considered unaffected in the range of 90–100%. Differences in mean absorbance were compared to the untreated control and considered significant when *p* < 0.05 (** *p* < 0.01, *** *p* < 0.001) according to the Student’s *t*-test.

**Figure 4 marinedrugs-22-00061-f004:**
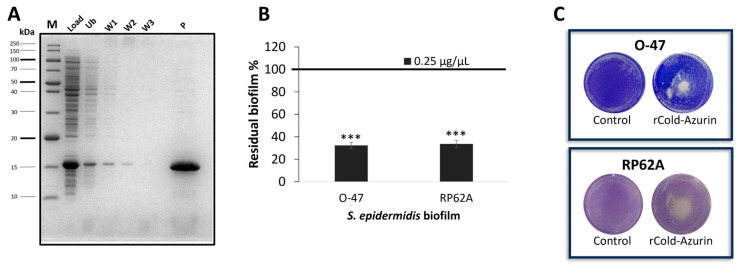
(**A**) SDS-PAGE 15% stained with Comassie blue, protein profile of the chromatography fractions, M: molecular weight marker, Load: periplasmic extract fraction from recombinant induced BL21DE3 cells, Ub: unbound fraction, W: fractions obtained during the washing step with MgSO_4_ (5 mM), P: eluted fraction with methanol. (**B**) Effect of *r*cold-azurin (0.25 µg µL^−1^) eluted from adsorption chromatography on *S. epidermidis* biofilm formation. Each data point represents the mean ± SD of four independent samples. The results are expressed as the percentage of biofilm formed in the presence of *r*cold-azurin compared to untreated bacteria (100%). Biofilm formation was considered unaffected in the range of 90–100%. Differences in mean absorbance were compared to the untreated control and considered significant when *p* < 0.05 (*** *p* < 0.001) according to the Student’s *t*-test. (**C**) Biofilm formation by *S. epidermidis* O-47 and RP62A in polystyrene 24-wells microtiter plate wells coated with PBS (Control) or *r*cold-azurin.

**Figure 5 marinedrugs-22-00061-f005:**
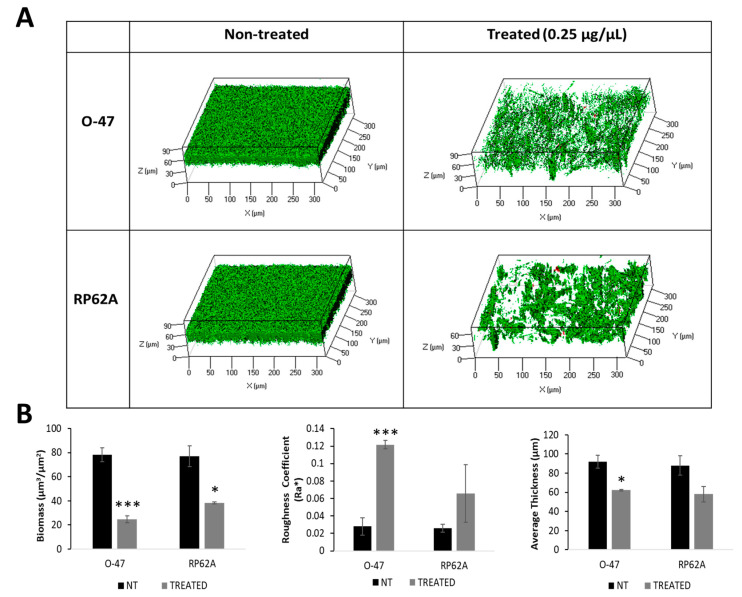
(**A**) CLSM analysis of *S. epidermidis* biofilms formed in the absence (non-treated) and presence (treated) of rcold-azurin (0.25 µg µL^−1^). (**B**) COMSTAT quantitative analysis of Biomass (µm^3^/µm^2^), Roughness Coefficient (Ra*), and Average thickness (µm) of untreated (NT) or treated (rcold-azurin) *S. epidermidis* biofilms. Differences in mean absorbance were compared to the untreated control and considered significant when *p* < 0.05 (* *p* < 0.05, *** *p* < 0.001) according to the Student’s *t*-test.

## Data Availability

The original data presented in the study are included in the article/Appendix A; further inquiries can be directed to the corresponding author.

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
