# Peer review of "Cold-Azurin, a New Antibiofilm Protein Produced by the Antarctic Marine Bacterium Pseudomonas sp. TAE6080"

_marinedrugs, 2024, doi:10.3390/md22020061_

Round 1
Reviewer 1 Report
Comments and Suggestions for Authors
Cold-Azurin a new antibiofilm protein produced by the Antarctic marine bacterium Pseudomonas sp. TAE6080
Caterina D’Angelo 1, Marika Trecca2, Andrea Carpentieri1, Marco Artini2, Laura Selan2, Maria Luisa Tutino1, 4, Rosanna Papa2, and Ermenegilda Parrilli 1*
In this manuscript, the authors study the antibiofilm properties of an azurin-like protein from Pseudomonas sp. TAE6080. In former publications, the authors have already shown the property of the supernatant of this bacterium and in the present manuscript, they show that the ability to inhibit the biofilm is due to an azurin-like protein. The anti-biofilm effect of cold-azurin produced in E. coli seems clear, but there are differences concerning the other extracts. Thus, the manuscript will improve if the authors complete some of the experiments or show other data from the experiments.
Major points:
In Figure 1, the photographs chosen by the authors do not reflect the percentage of biofilm inhibition. Please, choose other photographs or show a complete plate, or the OD data in supplemental material.
I missed a comparison of all the fractions tested for both strains in the same plate and experiment. Above all for the last two extracts E and OS. Although the properties of the heterologous azurin have been shown, the antibiofilm properties of flagellin should be tested, because they could explain some of the differences observed. Although OS extract is enriched in Azurin, there is still detectable flagellin. Photographs show more inhibition of extract E than extract OS.
The following sentences should be rephrased:
. Azurin is used by the bacterium as a multitarget weapon to avoid the entry of competitors and to eliminate foreign invaders in the host, in this way the bacterium preserves its own survival.
Bacteria try to eliminate invaders from the host or from their environment, whatever it is?
. Moreover, the studies on the use of Azurin from P. aeruginosa in different human pathologies [31,33,45,46], that….
This sentence can lead the reader to assume that it has already shown therapeutic properties in humans. The cited references indicate a potential use of the drug. Please, rephrase.
Minor points:
As a general comment, please revise the structure of the sentences. Some of them are too long and have no punctuation marks. In the references section, the microorganisms must be also in italics and correctly named.
May references 20 and 21 be not correct?
Comments on the Quality of English Language
The English is fine, but some sentences are too long. Please, rephrase.
Author Response
Dear Editor,
Please find enclosed the revised version of the manuscript entitled: Cold-Azurin a new antibiofilm protein produced by the Antarctic marine bacterium Pseudomonas sp. TAE6080 by Caterina D’Angelo, Marika Trecca, Andrea Carpentieri, Marco Artini, Laura Selan, Maria Luisa Tutino, Rosanna Papa and myself.
We are deeply obliged to the reviewers for the very careful evaluation of the manuscript and their valuable suggestions. Following the referees’ suggestions, the text and figures were modified. We hope that the revised version is now acceptable for publication in Marine Drugs.
We thank you again for your consideration and look forward to hearing from your office.
Yours sincerely,
Ermenegilda Parrilli
Reply to Reviewer comments:
For clarity, in this letter, each reply is reported in bold, and in the revised manuscript all changes have been highlighted by using red coloured text.
Comments and Suggestions for Authors
In this manuscript, the authors study the antibiofilm properties of an azurin-like protein from Pseudomonas sp. TAE6080. In former publications, the authors have already shown the property of the supernatant of this bacterium and in the present manuscript, they show that the ability to inhibit the biofilm is due to an azurin-like protein. The anti-biofilm effect of cold-azurin produced in E. coli seems clear, but there are differences concerning the other extracts. Thus, the manuscript will improve if the authors complete some of the experiments or show other data from the experiments.
Major points:
Figure 1, the photographs chosen by the authors do not reflect the percentage of biofilm inhibition. Please, choose other photographs or show a complete plate, or the OD data in supplemental material.
Re: We are deeply indebted to the Reviewer for having raised this point because it helped us to improve the text.
The authors are sorry for not having been sufficiently clear in the description of the results reported in Figure 1. Indeed, panels A and C display the results of an experiment different from that reported in panel B. The panel B is the outcome of an experiment aimed to evaluate the inhibition of the biofilm formation, while panels A and C show the result of a surface coating assay, a test of inhibiting the adhesion to the support. Briefly, in surface coating assay a volume of 5 μL of the tested sample is deposited onto the center of a well of a 24-well tissue-culture-treated polystyrene microtiter plate, then the plate is incubated at room temperature to allow the complete evaporation of the liquid in sterile conditions. After this step, the wells are filled with 1 mL of S. epidermidis cultures in exponential growth and incubated at 37 °C in static conditions. After 24 h, wells are rinsed with water and stained with 1 mL of 0.1% crystal violet. Stained wells were rinsed with water and dried, after that, they were photographed.
I missed a comparison of all the fractions tested for both strains in the same plate and experiment. Above all for the last two extracts E and OS. Although the properties of the heterologous azurin have been shown, the antibiofilm properties of flagellin should be tested, because they could explain some of the differences observed. Although OS extract is enriched in Azurin, there is still detectable flagellin. Photographs show more inhibition of extract E than extract OS.
Re: We are deeply obliged to the Reviewer for having raised these two points because it assisted us to improve the text.
What is shown in Figure 2A is the result of the coating assay on fraction E. For this fraction, the inhibition test of the biofilm formation was not reported in the old version of the paper, in the revised version we introduced this result. For fraction OS the results of both assays are shown.
Flagellin is extracellular protein, while azurin is usually present in the periplasm. Therefore, the activity found in the OS fraction can be mainly attributed to Cold-Azurin. Of course, it cannot be entirely ruled out that a low concentration of flagellin may be present in the OS fraction due to imperfect purification of the periplasmic extract. However, downstream of the data on recombinant protein production in E. coli, any potential and unlikely contribution of flagellin to the antibiofilm activity can be considered negligible.
The following sentences should be rephrased: Azurin is used by the bacterium as a multitarget weapon to avoid the entry of competitors and to eliminate foreign invaders in the host, in this way the bacterium preserves its own survival.
Bacteria try to eliminate invaders from the host or from their environment, whatever it is?
Re: According to the Referee’s suggestion, we modified the sentence to make it clearer. (L.327-328)
. Moreover, the studies on the use of Azurin from P. aeruginosa in different human pathologies [31,33,45,46], that….
This sentence can lead the reader to assume that it has already shown therapeutic properties in humans. The cited references indicate a potential use of the drug. Please, rephrase.
Re: According to the Referee’s suggestion, we modified the sentence to make it clearer (L345).
Minor points:
As a general comment, please revise the structure of the sentences. Some of them are too long and have no punctuation marks. In the references section, the microorganisms must be also in italics and correctly named.
Re: The text was modified as suggested.
May references 20 and 21 be not correct?
Re: The text was corrected as suggested.
Comments on the Quality of English Language
The English is fine, but some sentences are too long. Please, rephrase.
Re: The manuscript was carefully edited by an outside party
Reviewer 2 Report
Comments and Suggestions for Authors
1. The abstract needs more information about the research work rather than the background (first 6 lines).
2. Is the Cold azurin a novel peptide? What is the similarity with known azurin? Please provide the details in the abstract. Also, it is recommended to provide sequence alignment with known azurin.
3. It is suggested to do an ingel activity assay for recombinant peptide. Here is the reference. https://www.ncbi.nlm.nih.gov/pmc/articles/PMC4704198/
4. Please provide the control details in figure legends along with statistical analysis details.
5. References are missing in many places in material and methods.
6. What about the cytotoxicity? It is recommended to do cytotoxicity assays.
7. Statistical analysis section in material and methods.
8. Please include a separate section describing the limitations of the current study.
Author Response
Dear Editor,
Please find enclosed the revised version of the manuscript entitled: Cold-Azurin a new antibiofilm protein produced by the Antarctic marine bacterium Pseudomonas sp. TAE6080 by Caterina D’Angelo, Marika Trecca, Andrea Carpentieri, Marco Artini, Laura Selan, Maria Luisa Tutino, Rosanna Papa and myself.
We are deeply obliged to the reviewers for the very careful evaluation of the manuscript and their valuable suggestions. Following the referees’ suggestions, the text and figures were modified. We hope that the revised version is now acceptable for publication in Marine Drugs.
We thank you again for your consideration and look forward to hearing from your office.
Yours sincerely,
Ermenegilda Parrilli
Reply to Reviewer comments:
For clarity, in this letter, each reply is reported in bold, and in the revised manuscript all changes have been highlighted by using red colored text.
Comments and Suggestions for Authors
- The abstract needs more information about the research work rather than the background (first 6 lines).
Re: Following the referee's request, we modified the abstract
- Is the Cold azurin a novel peptide? What is the similarity with known azurin? Please provide the details in the abstract. Also, it is recommended to provide sequence alignment with known azurin.
Re: We are deeply indebted to the Reviewer for having raised this point because it helped us to improve the clarity of the paper. Indeed, the sequence alignment was just present in the old version of the paper (Figure S1) but probably was not so evident therefore we modified the text to underline this analysis.
- It is suggested to do an ingel activity assay for recombinant peptide. Here is the reference. https://www.ncbi.nlm.nih.gov/pmc/articles/PMC4704198/
Re: The assay reported in the study, although clever, serves to evaluate only the antimicrobial activity, and Cold-Azurin possesses only antibiofilm activity, without affecting bacterial vitality. Therefore, unfortunately, the suggested assay cannot apply to the protein under investigation in this work.
- Please provide the control details in figure legends along with statistical analysis details.
Re: The figure legends were modified as suggested.5. References are missing in many places in material and methods.
Re: The material and methods section was modified as suggested.
- What about the cytotoxicity? It is recommended to do cytotoxicity assays.
Re: Cytotoxicity will be assessed in the upcoming characterization studies of the protein. In the future, further experiments will be performed to evaluate the Cold-Azurin activity against other clinical pathogens and, in this case, we will evaluate the cytotoxicity on different eukaryotic cell lines. In any case, the data obtained on Azurin isolated from P. aeruginosa are encouraging, suggesting that Cold-Azurin, too, is likely non-toxic to eukaryotic cells.
- Statistical analysis section in material and methods.
Re: The material and methods section was modified as suggested.
- Please include a separate section describing the limitations of the current study.
Re: We thank the Reviewer for this comment the limitations of the current study were described in the discussion paragraph (lines 339-343)
Round 2
Reviewer 1 Report
Comments and Suggestions for Authors
The authors have addressed most of the required work and information. Thus, now it is suitable for publication.
Author Response
Dear Editor,
Please find enclosed the revised version of the manuscript entitled: Cold-Azurin a new antibiofilm protein produced by the Antarctic marine bacterium Pseudomonas sp. TAE6080 by Caterina D’Angelo, Marika Trecca, Andrea Carpentieri, Marco Artini, Laura Selan, Maria Luisa Tutino, Rosanna Papa and myself.
We are deeply obliged to you for the very careful evaluation of the manuscript and their valuable suggestions. We hope that the revised version is now acceptable for publication in Marine Drugs.
We thank you again for your consideration and look forward to hearing from your office.
Yours sincerely,
Ermenegilda Parrilli
Reply to Reviewer comments:
For clarity, in this letter, each reply is reported in bold, and in the revised manuscript all changes have been highlighted by using red coloured text.
Comments and Suggestions for Authors
Please, find below some minor recommendations:
- The title should be written as follows: “Cold-Azurin, a New Antibiofilm Protein Produced by the Antarctic Marine Bacterium Pseudomonas sp. TAE6080”
- Throughout the manuscript, including in the abstract:
- check the spelling of “LC-MS/MS” and “Pseudomonas sp. (“sp” should NOT be in italics)
- “cold-azurin” or “azurin” should not be capitalized, when it appears in a sentence. The same goes for “flagellin”.
- the spelling of the temperature should be homogenized (e.g. 56 °C)
In addition,
- Page 2, line 88: “…S. epidermidis O-47 [20], an agr mutant considered as a strong biofilm producer [21].…”
- Page 3, line 114: Add a point after “90–100%”
- Page 3, line 119: “As controls, the…”
- Page 5, line 168 and page 11, line 478: “in situ” should be in italics
- Page 5, lines 178-179: “We took advantage of this difference to understand which protein was responsible for the anti-biofilm and anti-adhesive activity.”
- Page 6, line 206: “…we produced…
- Page 7, line 248: delete one “%.”
- Page 7, line 263: …we reported …
- Page 7, line 270: “…capable of acting as an anti-adhesive and antibiofilm compound.”
- Page 8, lines 271-272: “…point and to collect…or subjected…”
- Page 8, lines 284-285: Suggestion “Several proteins have emerged as good candidates for biofilm treatment and prevention [26,27], but are generally hydrolytic enzymes. In our case, the two…”
- Page 8, line 288-289: “…copper-containing protein found mainly in the periplasmic space of various Gram-negative bacteria
- Page 8, lines 301-302: “…and in Pseudomonas aeruginosa PAO1, it is …”
- Page 8, lines 304-305: “… between Cu(I) and Cu(II), and also supports…by transferring electrons from…”
- Page 8, lines 307-309: “In an attempt to speculate that the antibiofilm activity of cold-azurin might be related to the Cu(2+)/Cu(+) reduction potentials of the type-1 copper site [31], we tested the putative antibiofilm activity of poxA1b laccase [32], a well-known blue copper protein oxidase (Figure S3)..”
- Page 8, line 315: “ this protein inhibits…”
- Page 9, lines 326-330: “…is that azurin is used …competitors into the host cell and to eliminate foreign invaders from the host organism. In this way, the bacteriul….the job of the immune system, which is made up of immunoglobulins….”
- Page 9, line 331: “…antibiofilm activity of cold-azurin”.
- Page 9, line 334: “Several S. epidermidis proteins play an important role in biofilm formation [40,41] [38,39].
- Page 9, lines 339-340: “…further experiments are required to assess whether the target of cold-azurin activity …
- Page 9, line 341-342: “In addition, future studies will aim at the….as a potential agent…”
- Page 9, lines 345-347: “…in various human pathologies [33,35,47,48], which also demonstrated its biocompatibility [34,49,50], are encouraging for the future use of cold-azurin in the treatment of human infections in combination…”
- Page 9, lines 357-358 and Page 11, line 478: “L-1” should be superscripted
- Page 10, line 379: “..0.1 and 0.001 OD600nm, respectively..”
- Page 11, line 430: “…in a stirred tank reactor…”
- Page 11, line 435: “…at 7,000 rpm.”
- Page 11, line 456: Add a space between “0.3” and “mL”
- Page 11, line 466: “To analyze…”
- Page 11, line 467: “…13,000 rpm…”
- Page 11, line 470: “An aliquot of 2 µL…”
- Page 11, line 476: Add a space between “15” and “kDa”
- Page 11, line 479: “…and 0.1 %...”
- Page 12, line 484: “…min-1” should be superscripted
- Page 12, line 487: “MIN” should not be capitalized
- Page 12, line 491: “m/z” should be in italics
- Page 12, line 506: Add a space between “30” and “g” and between “5” and “mL”
- Page 12, line 508, and line 511: “…(7,500 rpm…”
- Page 12, line 513: Delete the space between “was” and “suspended”
- Page 12, line 516: “…(6,500 rpm…”
- Page 12, line 530: “The products were electrophoresed on a 1,5 % agarose gel, purified using the High Pure…Kit…”
- Page 13, lines 534 and 540, 547: use “U.µL-1“ and “µg.mL-1”, respectively
- Page 13, lines 551 and 555: “…at 4,000 rpm” and “…at 8,000 rpm…”
- Page 13, lines 565 and 566: “S. epidermidis” should be in italics
- Page 13, line 583: Please check and correct “(μm3 μm−2) »
- Page 14, Check the “Acknowledgments section”
Re: We are deeply indebted to the Editor for her careful correction, the manuscript was accordingly modified
Reviewer 2 Report
Comments and Suggestions for Authors
The authors successfully responded to the reviewer's comments and updated the manuscript as well.
Author Response

(The authors gave the same response as above.)
